# Socioeconomic inequalities in reach, compliance and effectiveness of lifestyle interventions among workers: protocol for an individual participant data meta-analysis and equity-specific reanalysis

Karen M Oude Hengel,[1,2] Pieter Coenen,[3] Suzan J W Robroek,[1] Cecile R L Boot,[3] Allard J van der Beek,[3] Frank J Van Lenthe,[1,4] Alex Burdorf[1]

For numbered affiliations see end of article.

**Correspondence to**
Karen M Oude Hengel;
k.oudehengel@erasmusmc.nl

## ABSTRACT

**Introduction** Obesity and unhealthy behaviour are more prevalent among workers with a low compared with a high socioeconomic position (SEP), and thus contribute to socioeconomic health inequalities. The occupational setting is considered an important setting to address unhealthy behaviours due to the possibility to efficiently reach a large group of adults through worksite health promotion. This paper describes the rationale and design for an individual participant data (IPD) meta-analysis and a socioeconomic equity-specific reanalysis aiming to: (1) investigate socioeconomic differences in the effectiveness of interventions aimed at promoting healthy behaviour and preventing obesity, (2) examine socioeconomic differences in reach and compliance and (3) to investigate underlying factors affecting possible socioeconomic differences.

**Methods and analysis** A systematic search was conducted in electronic databases including Embase, Medline Ovid, Web of Science, Cochrane Central and Google Scholar as well as in grey literature and trial registries. Two researchers have independently selected a total of 34 relevant studies (from 88 articles). Responsible researchers of these eligible studies were asked to provide their study data and an assessment of the methodological criteria was done. The data of the intervention studies will be pooled for the IPD meta-analysis, whereas the socioeconomic equity-specific reanalysis will focus on each study separately, stratified for SEP. Both methods will be conducted to investigate socioeconomic differences in effectiveness, reach and compliance (research aims 1 and 2). For research aim 3, different factors, such as population characteristics, organisational work environment and intervention characteristics, will be investigated as possible moderators in the associations between SEP and effectiveness, reach and compliance.

**Ethics and dissemination** The Medical Ethical Committee of Erasmus MC declared that the Medical Research Involving Human Subjects Act does not apply to the meta-analyses. The findings will be disseminated through peer-reviewed publications and (inter)national conference presentations.

**Trial registration number** CRD42018099878.

### Strengths and limitations of this study

► The proposed meta-analyses will not only gain insight in socioeconomic inequalities in the effectiveness of worksite health promotion programmes but also on the socioeconomic differences in reach and compliance towards these interventions.

► The proposed meta-analyses will also investigate underlying factors (eg, intervention characteristics, individual characteristics and work-related factors) affecting possible socioeconomic differences in reach, compliance and effectiveness.

► The proposed analyses rely on the original data with the advantages of having enough statistical power, to standardise outcomes across studies and have access to additional factors.

► Due to large differences in legal conditions and the social context largely differ across countries, the proposed meta-analyses will be restricted to Dutch studies.

## INTRODUCTION

The prevalence of obesity is 2.5-fold higher among Dutch workers with a low as compared with a high socioeconomic position (SEP).[1] Unhealthy behaviours (eg, physical inactivity, smoking and unhealthy dietary intake) are also more prevalent among workers with low SEP.[2] Because unhealthy behaviour and obesity contribute substantially to socioeconomic health inequalities in the working population,[1] there is an urgent need for effective lifestyle interventions aimed at promoting healthy behaviour and/or preventing obesity among workers with low SEP. In order to reduce socioeconomic health inequalities, such preventive lifestyle interventions need to be targeted to workers with low SEP specifically, or target those risk factors that are more frequently present in workers with low

SEP.[3] WHO recommends delivering health promotion programmes at worksites[4] because it is a suitable setting to reach a large group of adults with low SEP and social support at work might be beneficial as well.

The effectiveness of lifestyle interventions among the working population has been studied in several systematic reviews,[5–12] considering a wide variety of interventions. Individual interventions mostly contain a cognitive and educational component, while environmental interventions often introduce healthy food in canteens or make adjustments in buildings to increase physical activity. The systematic reviews have shown positive effects on smoking cessation,[6] positive but small effects regarding improvement in dietary intake[7 10 11] and inconclusive effects on physical activity[7 9–11] and obesity.[5 7 8 10–12]

The question arises whether aforementioned worksite lifestyle interventions reduce or amplify socioeconomic health inequalities.[13] In a meta-analysis, Magnée et al[14] reanalysed Dutch lifestyle interventions among different SEP groups. None of the six worksite interventions decreased socioeconomic health inequalities between workers. In contrast, they found a larger intervention effect among workers with high compared with low SEP in two interventions with a cognitive component. Another systematic review confirmed that workplace interventions focusing on health education were ineffective in decreasing socioeconomic health inequalities,[15] although small positive effects for physical activity interventions targeted at workers with low SEP were found.

If studies aimed at preventing unhealthy lifestyle behaviour and obesity are ineffective among workers with low SEP, it raises the question of whether the intervention itself is not effective (theory failure) or whether these intervention are poorly implemented (programme failure).[16] Both theory failure and programme failure can be influenced by a wide range of different factors. Several systematic reviews provide evidence on the influence of intervention characteristics (eg, during working hours, weekly consults), work context (eg, social support, organisational structure) and study population (eg, younger age) on the effectiveness of healthy lifestyle interventions.[17–19] Insight in these underlying factors of (in)effectiveness as well as reach and compliance of lifestyle interventions is of eminent importance to develop and implement effective lifestyle interventions. However, it is unknown yet whether and which of these factors play a role in the differences between workers with low SEP compared with high SEP.

As the number of high-quality studies (eg, randomised controlled trials (RCTs)) evaluating the effectiveness of Dutch lifestyle interventions among workers has increased in the recent years, it is possible to provide knowledge on the effectiveness of lifestyle interventions and the reach and compliance to these interventions across workers from different socioeconomic groups in a meta-analysis. Since legal conditions and the social context largely differ across countries, comparisons across studies from different countries are difficult to interpret. Examples

of country-specific factors are the legislation on smoking ban at worksites (introduced in 2004 in the Netherlands) or provision of occupational health services in the Netherlands. In order to shed light on socioeconomic differences, it would be better to rule out aforementioned influences. Therefore, the proposed meta-analysis will be limited to studies conducted in the Netherlands and the results will therefore not be influenced by dissimilarities in the national context of social, economic and legal conditions. This paper describes the rationale and design for an individual participant data (IPD) meta-analysis and an equity-specific reanalysis of each intervention study separately. The first aim is to investigate socioeconomic differences in the effectiveness of Dutch interventions aimed at promoting healthy behaviour and preventing obesity. The secondary aim is to examine socioeconomic differences in reach and compliance to these interventions. Third, the meta-analysis aims to investigate which and to what extent factors influence differences in reach, compliance and effectiveness of the interventions.

## METHODS AND ANALYSIS

The current manuscript is prepared in accordance with the Preferred Reporting Items for Systematic review and Meta-Analysis Protocols (PRISMA-P) statement.[20] The described IPD meta-analysis has been a priori registered in PROSPERO (register number: CRD42018099878).

### Identification and selection of the studies

A systematic inventory was conducted to identify relevant published and unpublished Dutch intervention studies aimed at worksite promotion of healthy behaviour and prevention of obesity. First, a literature search was conducted in the electronic databases of Embase, Medline Ovid, Web of Science, Cochrane Central and Google scholar to obtain an overview of published studies. Search terms included a wide range of synonyms, both in subject headings and free-text words related to (1) healthy behaviour, (2) obesity, (3) intervention, (4) evaluation and (5) worker or worksite. These search terms were combined as follows: (#1 or #2) and #3 and #4 and #5. Moreover, the search was restricted to studies conducted in the Netherlands. No data restrictions were applied in the searches. Complete search strategies for the different electronic databases are added as online supplementary file.

Second, included studies retrieved from the search and three recently published systematic reviews and meta-analyses[14 21 22] were screened for additional relevant references. Third, in order to also identify relevant studies from the so-called grey literature and unpublished work, trial registers, major Dutch funding agencies and the intervention database of the National Institute of Public Health and the Environment (http://www.loket-gezondleven.nl) were checked for additional eligible studies. Lastly, researchers and experts in the field

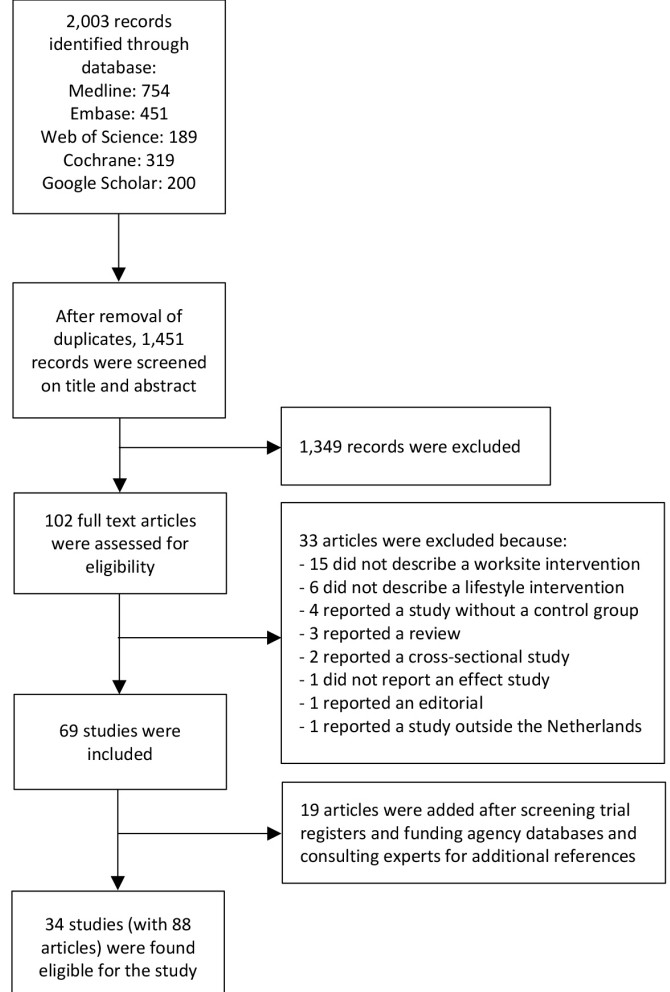

**Figure 1** Flow chart of study selection.

of occupational health were contacted for additional published or unpublished studies.

The literature search generated 1451 unique references that were screened on their title and/or abstract in April and May 2018 (figure 1). A total of 102 full-text articles were retrieved, of which 33 were excluded for various reasons. After adding 19 articles from other sources (from trial registries, funding agency databases and after consulting experts), 88 articles (from 34 studies) were found eligible for data extraction.

### Selection of the studies

Studies were included in case (1) it concerns a preventive intervention study aimed at promoting healthy behaviour or preventing obesity, (2) the intervention was targeted at workers, (3) the intervention was conducted in the Netherlands, (4) the study design met the methodological quality requirements (as described below) and (5) an indicator of SEP was measured.

Based on the classification of prevention, universal preventive, selective preventive and indicated preventive interventions were included.[23] Thereby, care-related interventions were excluded. Universal prevention includes lifestyle interventions that are targeted at a

general group of workers. Selective prevention includes interventions targeted at high-risk subpopulations identified as being at elevated risk for a disease (eg, cardiovascular diseases), whereas indicated prevention included interventions targeting workers who are individually identified as having an increased vulnerability for a disease but are not diagnosed yet.

Regarding the methodological quality of intervention studies, the most robust design would be an RCT. However, interventions in the occupational setting can be very complex and difficult to standardise, for instance, due to the multiple components and providers, the high turnover at worksites and multiple locations. Consequently, conducting an RCT is not always a feasible option.[24] Therefore, intervention studies were included if they were evaluated with an RCT, with at least a premeasurement and postmeasurement and a comparative reference group, or with one of the following alternative methods for analysing observational designs: propensity scores, methods of instrumental variables, multiple baseline design, interrupted time series, difference-in-difference and regression discontinuity design.[25]

Regarding the indicator of SEP, it is expected that educational level is included in most studies, and, thus, will be included as the primary indicator of SEP in the current study. According to the 1997 International Standard Classification of Education, the highest level of education completed will be categorised into low (preprimary, primary and lower secondary) or moderate/high (upper secondary and postsecondary) education. When this indicator is not included in the study, alternatives as income and occupational class will be used to define SEP.

Title and abstracts of the records generated from the searches were screened for eligibility by two researchers (PC and SJWR) in April and May 2018. Second, full texts of potential relevant records were obtained and screened. Disagreements were discussed and resolved during consensus meetings. If the researchers could not reach consensus, a third researcher (KMOH) was consulted in June 2018. Multiple publications of the same study were identified and linked for the data extraction.

### Data extracting and management

Since the proposed meta-analysis requires access to the original data, the project group sent an email of invitation to the principal investigator or first author of each eligible study. Reminders are sent and telephone contact is sought. When all these attempts fail, another author of the article (last, second, third, fourth, etc in order) was contacted. If the (principal) investigator expressed interest to share data for the current meta-analysis, a data sharing document was sent to explain the aims of the current study and which data are relevant for the current study. Reasons for refusal at any stage was and will be recorded.

A data extraction form, which has been created and pilot tested by the project group, is filled out for each participating study. Here, the following data is extracted from

each study: (1) study design, follow-up duration and loss-to-follow-up, (2) intervention type, content and setting, (3) characteristics of the participants, (4) primary and secondary outcomes including a specification of measurement method used and (5) the indicator measured for SEP. When available, process data including data about the reach and compliance as well as the work context are collected. Additional information about the study (eg, syntax, informed consent) and information about the ethical committee approval are documented.

Data were extracted based on the identified articles of each eligible study by one member of the project group (KMOH, PC or SJWR) and verified by another member (KMOH, PC or SJWR) from May to October 2018. The data extraction form is checked by the contact person of the specific study. During the data extraction and contact with the principal investigator, 9 of the 34 eligible studies were excluded because data were not available (n=4), primary outcomes were not included in the study (n=3), socioeconomic status was not measured (n=1) or investigators could not be reached (n=1).

After approval to release the data, researchers of the original study are asked to submit their data set including all potential characteristics measured before the intervention as well as the outcomes assessed during and after the intervention. The researcher of the specific study needs to anonymise the data to ensure that the data set will not contain any personal information that may identify an individual (eg, no birth date, no address information, no company name). Data can be transferred by the programme FileSender, which is especially developed to exchange research data between universities in compliance with the Dutch legislation, in any electronic format (eg, SPSS, STATA or Excel). Researchers of the current project group are available to assist when investigators ask for additional support to supply their data.

### Outcome measures
Primary outcome measures in the meta-analysis are the prevalence of obesity and/or healthy behaviour. Regarding obesity, outcomes related to body composition, such as body height, body weight and waist circumference, will be included by self-reported or objective assessed measures. Regarding healthy behaviour, outcomes related to physical activity and sedentary behaviour, smoking, alcohol consumption and dietary intake will be included. Physical activity and sedentary behaviour can either be assessed as self-reported time in various activities or by objective measures. Smoking can be assessed by current tobacco use per day (eg, number of cigarettes per day) or the smoking status of the participant (eg, non-smoker, pervious smoker, current smoker). Alcohol consumption is, for instance, the average number of glasses alcohol per week. Regarding dietary intake, outcomes such as fruit intake, vegetable intake and fat intake will be included, whereas the above-described primary outcomes are a direct measure of (un)healthy behaviour, effects on health outcomes such as blood pressure and cholesterol

level can be expected to result from a change in (un)healthy behaviour and will therefore be included as secondary outcomes.

### Harmonisation of the data
After the data sets have been transferred, the original data will be checked for completeness. For the intervention and, if available, control group, sample size, baseline characteristics of the study population on gender and age and observed mean preintervention and postintervention values of primary and secondary outcomes will be calculated and checked with the original publication. When discrepancies occur, the original researchers will be consulted.

Data will be harmonised for the IPD meta-analysis. The project group will formulate guidelines including the definition of cut-off points for primary and secondary outcomes measures to ensure that the reanalysis will be conducted as much as possible in a comparable way across the intervention studies. Therefore, a copy of the raw data of each trial will be recoded into a data file to match the specific variables for the proposed pooled statistical analyses. After all data sets have been merged into the new data file, the data will again be checked with the original raw data to ensure accuracy by a member of the project group. A codebook document will be drafted which includes the codes of the variables of the combined data set as well as each individual data set.

### Data quality assessment
Assessment of the quality of the study is of eminent importance as previous research has shown that studies with a lower methodological quality generally report larger effects of their interventions.[21 26] The methodological quality of the selected studies will be assessed using a nine-item checklist with methodological criteria as previously used in a meta-analysis of health promotion programmes at the worksite by Rongen et al[21] (table 1). The checklist is based on the guidelines in Cochrane Collaboration's tool for assessing risk of bias[27] and the checklist used by Verweij et al.[28] The checklist consists of criteria regarding randomisation procedure, blinding of participants, similarity of groups, compliance, loss to follow-up and intention-to-treat, adjustment for confounders, data collection method and follow-up. A study will be scored positive on a certain criterion if the quality criterion is met (1 point), negative if the quality criteria is not met (0 point) or unclear if the publication or additional information request by authors provides insufficient information to judge (also 0 points). In case of multiple outcomes or multiple interventions, publications can receive 0.5 points on criterion B and/or H if the criterion is true for only one outcome measure or intervention group. All studies will receive an overall methodological quality score based on the summation of positive scored items, with sum scores interpreted as excellent (8–9 points), good (4.5–7.5 points), fair (3–4 points) or poor (0–2.5

**Table 1** Methodological quality criteria

| Description | |
|---|---|
| A. Randomisation procedure | Positive if there was a clear description of the randomisation procedure and if the randomisation was adequately performed (ie, by a random selection of numbers or by a computer-generated list). |
| B. Blinding of participants | Positive if the participant was unaware of being assigned to the intervention group or control group |
| C. Similarity of groups | Positive if baseline characteristics of the comparison groups were comparable OR if there were important differences in potential confounders but these appropriately adjusted for in the analysis. |
| D. Compliance | Positive if participants attended the intervention satisfactorily according to the opinion of the reviewers |
| E. Loss to follow-up | Positive if the percentage of dropouts during the study period did not exceed 20% for short-term follow-up (=3 months) OR 30% for long-term follow-up (>3 months) |
| F. Intention-to-treat | Positive if an intention-to-treat analysis was performed for the outcome variable |
| G. Controlled for confounders | Positive if the analysis was controlled for potential confounders |
| H. Data collection method | Positive if data collection tools shown to be credible (eg, shown to be valid and reliable in published research, OR in a pilot study, OR taken from a published national survey, OR recognised as an acceptable measure (such as biochemical measures of smoking)). |
| I. Follow-up | Positive if follow-up was at least 6 months |

points). Two members of the project group (combinations of PC, SJWR and KMOH) will assess and check the methodological quality.

### Statistical analyses

To address the specific research aims, two statistical approaches will be conducted, namely (1) an IPD meta-analysis with SEP as interaction term and (2) an socioeconomic equity-specific reanalysis of each intervention study by stratifying for SEP and visualised by a harvest plot.

#### Aim 1: socioeconomic differences in effectiveness of interventions

Regarding the IPD meta-analysis, either linear or logistic mixed modelling will be conducted using a three-level structure (worker, department/company/occupational physician and study) to take the clustering of workers within studies into account. According to the intention-to-treat principle, regression analyses will be conducted with the postintervention value as the outcome, adjusted for the baseline values and for relevant confounders. In forest plots, effect sizes will be reported depicting each individual study, as well as pooled effect sizes. Heterogeneity among studies will be assessed using $I^2$ statistics and visual inspection of the forest plots will be performed. A sensitivity analysis will be performed to assess the effect of individual study findings on the pooled results. To assess the effect of methodological quality on the study outcomes, a cumulative meta-analysis will be conducted, starting with studies of high methodological quality up to those of low methodological quality. Funnel plots will be generated to assess publication bias (through visual inspection).

To address research aim 1, socioeconomic differences in overall effectiveness on primary (and possibly secondary)

outcomes will be examined by adding intervention, SEP and its interaction term 'SEP*intervention' into the regression model. Stratified analyses will be performed for workers with low and high SEP, respectively. Within these stratified analysis, the intervention effect over time will be estimated by considering the interaction term 'time*intervention'.

Aforementioned standard IPD meta-analytical technique may not be possible for all study outcomes, because of, for instance, the presence of heterogeneity in the assessment and timing of the outcomes. Therefore, a socioeconomic equity-specific reanalysis of each intervention study separately and visualised by a harvest plot will be conducted to address research aim 1. The aim of the equity-specific reanalysis is to compare the effectiveness of lifestyle interventions among workers with low and high SEP within the original study. The harvest plot was developed earlier for the synthesis of evidence of socioeconomic differential effects of interventions and can provide visualisation by combining results from different study designs and outcomes.[29] Moreover, this graphical form is able to demonstrate both the outcomes measures and quality of the study.[30] Harvest plots are therefore not only helpful in showing the direction of the intervention effects in relation to the study quality, but also in identifying major evidence gaps.[12] For each intervention study, stratified analyses will be conducted based on mixed models to estimate overall intervention effects. Thereafter, the effects of the interventions at different time points will be estimated by adding time and the interaction time*intervention to the model.

Because of an expected lack of statistical power within particular studies, as the studies were originally not designed for subgroup analyses, the differential

effectiveness of each intervention will be defined based on three different aspects: (1) existence of significant interaction effects, (2) point estimates of the effect in one subgroup being outside the 95% CI around the estimated effect in the other subgroup and (3) differences in significance ($p < 0.05$) of separate subgroup effects. For each study, the decision of differential effectiveness will be made by the current project team, and at least one researcher involved in the original effect evaluation will subsequently be asked to check and approve this decision. If a study consists of multiple intervention groups, all will be included in the decision on differential effectiveness.

### Aim 2: socioeconomic differences in reach and compliance to interventions

To address the socioeconomic differences in reach and compliance of intervention between workers with low and high SEP, the same statistical procedures of the IPD meta-analysis used to address research aim one will be performed, while using reach and compliance as dependent variables. By adding intervention, SEP and its interaction term 'SEP*intervention' into the regression model, the socioeconomic differences in reach and compliance can be assessed.

### Aim 3: factors influencing the effectiveness, reach and compliance

The third aim will be to gain insight into which factors possibly moderate the association between SEP and reach, compliance and effectiveness. As described previously, these data will be derived for each study during the data extraction stage.

The statistical procedures from the IPD meta-analysis used to address research aims 1 and 2 will be expanded by adding interaction terms for 'intervention' and several factors as possible moderator. Here, the following individual factors will be considered: organisational work environment (eg, company size), working conditions (probably mostly unpublished, for instance, job autonomy, job control, contract type, shift work) and population characteristics (eg, age, job type and gender). If an interaction effect shows to be statistically significant, stratified analyses will be carried out and the results will be presented accordingly.

The equity-specific reanalysis of each intervention study separately and visualised by a harvest plot will also be conducted to address research aim 3. For the stratified subgroup analyses of each intervention study separately, the harvest plots will be expanded with intervention characteristics, reach and compliance. For example, based on the content of the intervention, the study will be assigned to one of the categories (individual, environmental physical, environmental social or a combination). These categories reflect the current debate as to whether interventions with an environmental approach (eg, healthier food supply in canteens) are more effective among lower educated groups than individually (eg, personal education) oriented interventions.[31] If differences exist, harvest plots will be presented separately for these factors.

### Timeline

A search for eligible studies was conducted (April to May 2018), and data extraction and consultation with researchers was done from May to November 2018. Hereafter, 25 studies remained eligible for data collection. As of November 2018, data of a total of 13 studies have already been collected and harmonised. Researchers of the remaining 12 studies will be contacted again to ask for permission to use their data. Data collection will be completed in January 2019, with exception of two studies that are still under study by the principal investigators. Merging of the data sets and preparation of analysis scripts will be conducted in the period August 2018 to January 2019 to ensure that analyses can start when all data are collected by the end of January 2019. The project team will analyse all data from January 2019 onwards and it is expected to submit the first scientific paper to an international journal in the autumn of 2019.

### Patient and public involvement

Patients (ie, workers) and the public were not involved in development of the research question and outcome measures, nor the study design. The study does not involve recruitment of participants, and participants were not involved in the conduct of the study. The advisory board, consisting of representatives of workers and experts in the field of occupational health, will be consulted for the dissemination of the project towards the target group.

### Ethics and dissemination

The Medical Ethical Committee of Erasmus MC Rotterdam declared that the Medical Research Involving Human Subjects Act does not apply to the proposed meta-analyses. The purpose of the study is to offer insight into how to develop effective lifestyle interventions for workers with a low SEP and how to implement and deliver these interventions to those workers in order to reduce socioeconomic health inequalities. The findings of the proposed meta-analyses will therefore be disseminated through one or more peer-reviewed publications according to the PRISMA-P guidelines and presentations at (inter)national conferences but also through factsheets and infographics for the target group (eg, companies, workers with low SEP, sector organisations).

## DISCUSSION

The meta-analysis as described in this protocol is, to our knowledge, the first study that addresses socioeconomic differences in reach, compliance and effectiveness of lifestyle intervention studies among workers. As interventions seemed to be more easily adopted by workers with high SEP[3] and participation of workers with low SEP is generally low in lifestyle interventions,[14] the underlying factors explaining the (in)effectiveness, reach and compliance among workers will also be investigated in this meta-analysis.

A strength of this study is the research methodology. The meta-analysis will adopt the methodology of collecting original data instead of using data extracted from publications as in a conventional meta-analysis. This bears the advantages to (1) have enough statistical power for stratified analyses on original studies that were not designed with the explicit goal of investigating socioeconomic differences in intervention effects, (2) standardise outcomes across studies, for instance, by using equal cut-off points on physical activity and food intake when the original studies used different cut-off points and (3) have access to additional factors, such as intervention characteristics, study population and work context, that have not been reported in the publications. As earlier research found that higher quality studies showed lower effects on healthy behaviour and obesity prevention,[21 26] the quality of the studies will be taken into account by a cumulative meta-analysis in the IPD meta-analysis and by the harvest plot in the equity-specific reanalysis.

Some limitations should be considered as well. The proposed meta-analysis will only concern the Dutch work context because cross-national comparisons are hampered by large differences in legal conditions and the social context (eg, smoking ban and provision of occupational health services). Moreover, by restricting to the Dutch setting, studies from the grey literature and unpublished work could be included as well. This will provide an unbiased selection as scientific, peer-reviewed publication of the results is not a criterion for inclusion (ie, less publication bias). As high quality studies of other countries are excluded, other researchers will be encouraged to use this protocol paper as an example to report intervention effects for different socioeconomic groups in other (Western) countries facing socioeconomic inequalities in obesity and unhealthy behaviour. A second potential limitation is that, by definition, the meta-analysis will rely on the variables assessed in previously conducted intervention studies. If only a limited number of studies can be included with differences in characteristics and variables, then it is possible that not all outcomes and underlying factors of interest can be examined in the current meta-analysis. Third, it is expected that not all studies have well documented their study information regarding reach and compliance or that this information may differ largely.

## CONCLUSION

This protocol describes the design of the IPD meta-analysis and equity-specific reanalysis aiming to provide insight into the effectiveness of lifestyle interventions as well as the reach and compliance towards these interventions, and their underlying factors. Thereby, the meta-analysis may contribute to answering the urgent call of researchers, policymakers and employers regarding which and how workplace lifestyle interventions should be implemented to reduce socioeconomic health inequalities in the working population.

**Author affiliations**
[1]Department of Public Health, Erasmus University Medical Center, Rotterdam, The Netherlands
[2]Department of Work Health Technology, Netherlands Organization for Applied Scrientific Research TNO, Leiden, The Netherlands
[3]Department of Public and Occupational Health, Amsterdam Public Health research institute, VU University Medical Center, Amsterdam, The Netherlands
[4]Department of Human Geography and Spatial Planning, Utrecht University, Utrecht, The Netherlands

**Correction notice** This article has been corrected since it first published online. The open access licence type has been amended.

**Contributors** KMOH and PC drafted the manuscript. SJWR, CRLB, AJvdB, AB and FJVL provided intellectual input and critically reviewed the manuscript. All authors read and approved the final manuscript.

**Funding** This study is funded by The Netherlands Organization for Health Research and Development (ZonMw; dossier number 50-53115-98-009).

**Disclaimer** The funder had no influence on the design of the study, data collection or writing of the manuscript.

**Competing interests** None declared.

**Patient consent for publication** Not required.

**Provenance and peer review** Not commissioned; externally peer reviewed.

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
