## [Reviewer comments · BMJ Open]

This paper was submitted to a another journal from BMJ but declined for publication following peer review. The authors addressed the reviewers' comments and submitted the revised paper to BMJ Open. The paper was subsequently accepted for publication at BMJ Open.

(This paper received three reviews from its previous journal but only two reviewers agreed to published their review.)

ARTICLE DETAILS

TITLE (PROVISIONAL)	Socioeconomic inequalities in reach, compliance and effectiveness of lifestyle interventions among workers: protocol for an individual participant data meta-analysis and equity-specific re-analysis
AUTHORS	Oude Hengel, Karen; Coenen, Pieter; Robroek, Suzan; Boot, Cecile R. L.; van der Beek, Allard; Van Lenthe, Frank J.; Burdorf, Alex

VERSION 1 – REVIEW

REVIEWER	Johannes Siegrist Heinrich-Heine-University Duesseldorf Germany
REVIEW RETURNED	21-Aug-2018

GENERAL COMMENTS	Overall, this is a carefully prepared manuscript on an important question, demonstrating excellent knowledge of state of art and methodological standards. With its high quality it meets publication Standards of the Journal. However, several minor queries deserve attention: 1. In the Introduction and Discussion authors justify the restriction to Dutch studies by referring to country-specific legal, policy and contextual factors that might complicate the analysis. In my view, these reasons should be explained more convincingly, because the trade-off is the exclusion of more high quality studies from other countries (e.g.RCTs, where these confounding factors seem less problematic).2. On p.9 authors should define a minimum of studies needed to ensure statistical power, in case several original study authors are not willing to release their data.3. On p. 10, Primary and secondary outcomes should be defined (this occurs on p.12), and the role of secondary outcomes should be clarified (e.g. cardiovascular risk factors are used to define the different intervention groups).4. Concerning data quality assessment and figure 1, authors may consider to include the Navigation Guide developed by Lam et al. 2016 to assess risk of bias.
---

	5. There is a clear risk of multiple testing in the proposed strategy of statistical analysis. For instance, effectiveness will be evaluated with at least 7 indicators. Authors should explain how they cope with multiple testing or how they reduce the number of analyses.
--	--

VERSION 1 – AUTHOR RESPONSE

Reviewers' Reports

Overall, this is a carefully prepared manuscript on an important question, demonstrating excellent knowledge of state of art and methodological standards. With its high quality it meets publication Standards of the Journal.

1. In the Introduction and Discussion authors justify the restriction to Dutch studies by referring to country-specific legal, policy and contextual factors that might complicate the analysis. In my view, these reasons should be explained more convincingly, because the trade-off is the exclusion of more high quality studies from other countries (e.g. RCTs, where these confounding factors seem less problematic).

We restricted our study search indeed to the Netherlands because of country-specific legal, policy and contextual factors that might complicate the analyses and the interpretation of the results. For instance, the introduction of a smoking ban at the worksites and provision of occupational health services are country specific. By restricting the search to the Netherlands, it is possible to compare the results from the different studies in meta-analyses with a similar context.

We have added the following sentence to the Introduction:

Examples of country specific factors are the legislation on smoking ban at worksites (introduced in 2004 in the Netherlands) or provision of occupational health services in the Netherlands.

In the Discussion section, we described also that by restricting to Dutch studies, we are able to search in the grey literature and unpublished work through trial registers, major Dutch funding agencies and the intervention databases without any language restrictions. We added a limitation to our restriction to the Dutch studies.

We have added the following to the Discussion:

The proposed meta-analysis will only concern the Dutch work context because cross-national comparisons are hampered by large differences in legal conditions and the social context (e.g. legislation on smoking ban and provision of occupational health services). Moreover, by restricting to the Dutch setting, studies from the grey literature and unpublished work could be included as well. This will provide a relatively unbiased selection as scientific, peer-reviewed publication of the results is not a criterion for inclusion (i.e. less publication bias).

2. On p.9 authors should define a minimum of studies needed to ensure statistical power, in case several original study authors are not willing to release their data.

Since submission of the current manuscript, all principal investigators of the 34 eligible studies have been contacted. Up to now, 13 researchers released their data to our project team. Therefore, no statistical power issues are expected to occur.

3. On p. 10, primary and secondary outcomes should be defined (this occurs on p.12), and the role of secondary outcomes should be clarified (e.g. cardiovascular risk factors are used to define the different intervention groups).

We agree that defining the primary and secondary outcomes need to be clarified earlier in the manuscript and have therefore moved this paragraph to page 10, before data harmonization is described. We would like to mention that secondary outcomes are not used to define the different interventions groups as mentioned by the reviewer. To clarify, the primary outcomes are direct measures of (un)healthy behavior (e.g. physical activity, nutrition and alcohol intake, smoking), whereas the secondary outcomes are health outcomes which might improve as a result of (a combination of) a change in healthy behavior.

We have added the following sentence in the manuscript:

Whereas the above described primary outcomes are a direct measure of (un)healthy behaviour, effects on health outcomes such as blood pressure, cholesterol level and cardiovascular risk profile can be expected to result from a change in (un)healthy behaviour and will therefore be included as secondary outcomes.

4. Concerning data quality assessment and figure 1, authors may consider to include the Navigation Guide developed by Lam et al. 2016 to assess risk of bias.

We considered the reviewer's suggestion to include the navigation guide for the data quality assessment. This guide incorporated evidence from clinical and environmental health sciences and provides therewith an approach for evaluating and integrating human and nonhuman evidence streams (Lam, 2014). After carefully reading the article, the basics of this guide are similar as used in the current study (i.e. risk of bias, strength of evidence and quality of evidence). The main strength of the proposed methodology by the reviewer is the possibility for integrating different sources of data (e.g. human data and animal data). As our meta-analyses includes only data from intervention studies among workers, the sources of the data in the current study are comparable. Therefore, we decided not to strictly follow the navigation guide on data quality assessment, but its key principles have been adopted in the evaluation that will be used in the current study.

5. There is a clear risk of multiple testing in the proposed strategy of statistical analysis. For instance, effectiveness will be evaluated with at least 7 indicators. Authors should explain how they cope with multiple testing or how they reduce the number of analyses.

We agree with the reviewer that there may be a risk of multiple testing. As mentioned by the reviewer, the effectiveness will be evaluated with several indicators of (un)healthy behavior and health outcomes. Thereby, all analyses will test for interaction with educational level.

To respond on the first concern about the effectiveness, we will follow the analyses as done in the original studies with regard to aim 1 and 2 (effectiveness and compliance). As these intervention studies mostly evaluate several different behaviors and health outcomes, as part of our meta-analyses we will conduct a large amount of analyses as well. However, the risk of multiple testing will not influence the main results, since we do not focus on associations reported in a single study (high risk of artificial finding with multiple testing), but we conduct a pooled analysis across different studies (thereby eliminating p-value driven reporting in single studies). A second strategy is that we will carefully link the content of the intervention to the outcome measure, i.e. an intervention aimed at quitting smoking will not be evaluated for its effect on physical activity. Besides, our conclusion will not be based on the results of one study, but will be based on the underlying patterns found across studies.

VERSION 2 – REVIEW

REVIEWER	Johannes Siegrist Faculty of Medicine, Heinrich-Heine-University Düsseldorf, Germany
REVIEW RETURNED	21-Nov-2018
GENERAL COMMENTS	This revised Version of the manuscript has addressed Major comments raised in previous Reviews. Specifically, more detailed Information was given with regard to data extraction, outcome measure, timeline and - in the Discussion section - justification of restriction of Analysis to Dutch studies. Based on These improvements I recommend publication of this contribution in its updated Version.